# Efficacy of Different Dosing Regimens of IgE Targeted Biologic Omalizumab for Chronic Spontaneous Urticaria in Adult and Pediatric Populations: A Meta-Analysis

**DOI:** 10.3390/healthcare10122579

**Published:** 2022-12-19

**Authors:** Humayun Manzoor, Foha Razi, Amina Rasheed, Zouina Sarfraz, Azza Sarfraz, Karla Robles-Velasco, Miguel Felix, Ivan Cherrez-Ojeda

**Affiliations:** 1Department of Research, Bahria University Medical & Dental College, Karachi 75260, Pakistan; 2Department of Research, Nishtar Medical University, Multan 60000, Pakistan; 3Department of Research, Sahiwal Medical College, Sahiwal 57040, Pakistan; 4Department of Research and Publications, Fatima Jinnah Medical University, Lahore 54000, Pakistan; 5Department of Pediatrics and Child Health, The Aga Khan University, Karachi 74000, Pakistan; 6Allergy, Immunology & Pulmonary Medicine, Universidad Espíritu Santo, Samborondón 092301, Ecuador; 7Department of Medicine, New York City Health + Hospitals, Lincoln Hospital, Bronx, NY 10451, USA

**Keywords:** omalizumab, biologics, IgE, chronic spontaneous urticaria, dosing regimens, meta-analysis

## Abstract

Chronic urticaria is a debilitating skin condition that is defined as itchy hives at least twice a week and lasting for six or more weeks, with or without angioedema. Chronic spontaneous urticaria (CSU) is a form of disease that is witnessed in two-thirds of those with chronic urticaria. This meta-analysis explores the efficacy of differential dosages of omalizumab for outcomes of weekly itching scores, weekly wheal scores, urticarial assessment score 7 (UAS7), and responder rates. Adhering to PRISMA Statement 2020 guidelines, a systematic search of PubMed/MEDLINE, Scopus, Embase, and Web of Science was conducted until 15 September 2022. A combination of the following keywords was used: omalizumab and chronic urticaria. Data comprising clinical trial ID, name, author/year, country, dosage and time of intervention, inclusion criteria, mean age, female gender, and racial grouping information were obtained. The meta-analytical outcomes were analyzed in RevMan 5.4. The risk-of-bias assessment was conducted using version 2 of the Cochrane risk-of-bias tool for randomized trials (RoB 2). A total of 10 trials comprising 1705 patients with CSU were included. Notably, 1162 belonged to the intervention group, while 543 were controls. A total of 70.4% of the participants were female in the intervention group, while 65.6% of them were female in the control group. The overall mean age was 38.64 ± 10.66 years. Weekly itch score outcomes were most notable with 150 mg dosage (Cohen’s d = −2.6, 95% CI = −4.75, −0.46, *p* = 0.02). The weekly wheal score outcomes had the largest effect size with 300 mg dosage (Cohen’s d = −1.45, 95% CI = −2.2, −0.69, *p* = 0.0002). For UAS7 outcomes, the largest effect size was yielded with 150 mg dosage (Cohen’s d = −6.92, 95% CI: −10.38, −3.47, *p* < 0.0001). The response rate to omalizumab had a likelihood of being higher with 300 mg of intervention compared to placebo (OR = 8.65, 95% CI = 4.42, 16.93, *p* < 0.0001). Well-rounded urticarial symptom resolution was observed with 150 mg and 300 mg dosages of omalizumab. Improvement of UAS7 was more comparable with 150 mg dosage, whereas the chance of response to treatment was higher with 300 mg dosage. Our findings support omalizumab as an effective intervention for adult and pediatric populations that are resistant to many therapies, including high-dose H1-antihistamines.

## 1. Introduction

Chronic urticaria (CU) is a debilitating skin condition defined as itchy hives at least twice a week and lasting for 6 or more weeks, with or without angioedema, and it affects 1–2% of the population [1]. CU is associated with a compromised health-related quality of life among patients, including psychiatric comorbidity, e.g., depression and/or anxiety, and disrupted daily routine activities [2]. The duration ranges from 1 to >5 years in more than 10% of patients [3,4]. A form of CU is idiopathic urticaria (CIU), also known as chronic spontaneous urticaria (CSU), which has no known external triggers, although an autoimmune basis is being explored widely [5]. Nearly two-thirds of the patients with CU are diagnosed with CSU [6]. The first line of treatment is H1 antihistamines among patients with CSU (doses can exceed up to 4 times). However, as much as 55–60% of patients will still suffer from CSU-related symptomatology even at higher doses [7,8]. Cutaneous mast cell release of mediators, primarily histamine, are known to be major effector cells in most types of urticaria, although other cells can also be involved. Immunoglobulin E (IgE) and basophils are suspected to stimulate the release of histamine in patients with CSU [9]. In treatment-resistant CSU, the role of functional autoantibodies against immunoglobulin E (IgE) receptor triggered by circulating IgE against autoantigens, IgG against FcεRI, or by IgG against IgE itself has been observed in the sera of patients with CSU [10,11,12,13].

Recent advances have identified omalizumab, which is a recombinant humanized monoclonal IgG anti-IgE antibody, a biologic initially indicated for allergy-induced asthma, as it may suppress mast-cell activity and reduce symptomatology in patients with severe disease. Up to 45% of H1-antihistamine unresponsive patients with CSU have responded to omalizumab, thereby promoting efforts to establish omalizumab’s efficacy in CSU [2]. Omalizumab targets IgE when bound to its high-affinity receptor, FcεRI [14]. The biological agent is also able to accelerate the dissociation of pre-bound IgE in basophils [15]. Recent studies have found the activity of omalizumab in basophils and dermal mast cells when given in physiological dose ranges [15]. However, the efficacy of omalizumab has been reported in both autoimmune and non-autoimmune CSU cases [16]. In an attempt to better understand the appropriate dosing of omalizumab in CSU, the latest evidence on the efficacy of different doses of omalizumab across clinical trials has not been analyzed. The following meta-analysis explores the efficacy of different doses of omalizumab for improvement in (1) weekly itching scores, (2) weekly wheal score outcomes, (3) urticarial assessment score 7 (UAS7), and (4) responder rates.

## 2. Materials and Methods

### 2.1. Search Strategy and Inclusion Criteria

A systematic search of PubMed/MEDLINE, Embase, Scopus, and Web of Science databases was conducted from inception to 15 September 2022, using the following keywords: omalizumab and chronic urticaria. The full keyword string is appended in Appendix A, ST 1. An additional search of ClinicalTrials.Gov (accessed on 16 October 2022) was conducted to obtain data from completed trials that were not published in the form of a journal article. By using this approach, two reviewers (Z.S. and A.S.) reviewed the studies independently against the inclusion criteria. The titles and abstracts of shortlisted studies from the enlisted databases and an additional search of ClinicalTrials.Gov were screened independently by two reviewers (Z.S. and A.S.). During the screening phase, the reference lists of the studies were also assessed to ensure no data were omitted (Umbrella methodology). In case of any disagreements, the third reviewer (I.C.-O.) resolved them to reach a consensus. Cohen’s coefficient of the inter-reviewer agreement was computed.

Patients who were diagnosed with CU and who were being intervened with omalizumab in a randomized controlled trial setting were included. Any case series, case reports, cohorts (retrospective or prospective), systematic reviews, meta-analyses, letters, and brief reports were omitted. The outcome measures included weekly itch scores, weekly wheal scores, urticaria assessment score 7 (UAS7), and responder rates. These outcomes were segregated based on three dosage methods, including 75 mg, 150 mg, and 300 mg.

### 2.2. Data Extraction (Selection and Coding)

Two reviewers independently extracted the data obtained from the studies into a spreadsheet. The third reviewer was present for any disagreements. The pair of reviewers identified the (1) clinical trial ID and name, (2) author and year, (3) country of origin, (4) dosage and period of intervention, (5) inclusion criteria, (6) mean age of enrolled participants, (7) female gender in Intervention Group (IG) versus Control Group (CG), and (8) race (IG vs. CG). These are enlisted in Table 1. The meta-analytical outcomes were inputted as well into a spreadsheet with the following data for IG vs. CG: weekly itch score, weekly wheal score, responders to therapy, and UAS7 outcomes. The data sheet is appended in Appendix A, ST 2.

The individual study data were entered in a presentable format during the data extract phase, and a clinical relevance assessment was additionally conducted. The bibliographic entries were recorded in the data software EndNote X9 (Clarivate, London, UK), and all duplicates were omitted and managed using the software. The referencing software utilized for this study was Mendeley (Elsevier, Amsterdam, The Netherlands), where all cited studies were managed. The Kappa score, which is the inter-rater reliability measure of agreement, was computed using the Statistical Package for Social Sciences (SPSS, v24).

### 2.3. Data Analysis

The meta-analysis was conducted using Review Manager (RevMan) 5.4. A quantitative analytical methodology was applied to ascertain the difference in the pre-specified outcomes post-omalizumab dose-based intervention. For continuous variables, namely the mean difference (MD) and the standardized mean difference (SMD) reported as Cohen’s d, applying 95% confidence intervals (CI) was reported. The minimum requirement to conduct the meta-analysis for an outcome was at least two or more trials reporting on the same outcome measure. A funnel plot was generated; Cochrane’s handbook defines it as a simple scatter plot of the intervention effect estimates from individual studies against the measure of each study’s size or precision. The heterogeneity across the included studies was assessed using the I^2^ index and the χ²-based Q test.

### 2.4. Risk of Bias (Quality) Assessment

The included RCTs were assessed for risk of bias using version 2 of the Cochrane risk-of-bias tool for randomized trials (RoB 2). The RoB 2 assessment comprises five domains, as follows: (1) bias arising due to the randomization process, (2) bias arising due to deviations from the intended interventions, (3) bias occurring due to missing outcome data, (4) bias arising during the measurement of the outcome, and (5) bias in the selection of the reported result. Domain-level judgments concerning the risk of bias were classified and reported as follows: (1) low risk of bias, (2) some concerns, and (3) high risk of bias. Both the traffic light plot and the weighted summary plot of the domain-level classification are illustrated in Section 3.6: risk of bias synthesis.

### 2.5. Protocol Registration and Funding Role

The protocol of this meta-analysis was registered with PROSPERO (PROSPERO 2022 CRD42022368256). No external funding was obtained.

## 3. Results

Of the 3492 studies and records identified from the databases and ClinicalTrials.Gov, all of them were screened. Post-screening of the titles and abstracts/summaries, 3286 entries were excluded as they did not fulfill the inclusion criteria, and 78 studies were assessed with full-text scanning for eligibility. On full-text appraisal, 68 studies were removed for multiple reasons, and 10 trials were included in this meta-analysis. The PRISMA flowchart for the study selection process is illustrated in Figure 1. Kappa’s inter-reviewer agreement score was calculated to be 0.91.

### 3.1. Overview of the Included Studies

In this meta-analysis, we included 10 trials, pooling in a total of 1705 patients with CSU. Of these, 1162 belonged to the intervention group and 543 belonged to the control group. Notably, 70.4% of participants in the intervention group were female, whereas 65.6% of participants in the control group were female. Both pediatric and adult populations were included, with an overall mean age of 38.64 ± 10.66 years computed for the entire sample. All participants in this meta-analysis were formally diagnosed with CU and were being treated with omalizumab ranging between 75 mg to 600 mg. The majority of interventions were conducted for 24 weeks, with the minimum total intervention time spanning 12 weeks. The characteristics of the included studies are listed in Table 1.

### 3.2. Weekly Itch Score Outcomes

Omalizumab 75, 300, or 600 mg subcutaneously for a total of 1 dose and followed for 24 weeks.

When comparing the 75 mg dosage of omalizumab to the placebo, the mean difference for the weekly itch score was computed as MD: −1.70 (95% CI: −3.09, −0.31). There was limited heterogeneity (I^2^ = 20%, *p* = 0.02) (Figure 2A). On computing the SMD, the values for Cohen’s d were as follows: −0.29 (95% CI: −0.54, −0.04). While the effect size was small, it was in favor of the intervention (I^2^ = 25%, *p* = 0.02) (Figure 2B).

On comparing the 150 mg dosage to the placebo of omalizumab, the mean difference in the weekly itch score was computed as MD: −2.98 (95% CI: −4.09, −1.88) (Figure 3A). There was high heterogeneity (I² = 97%, *p* < 0.0001) (Figure 3A). A large effect size was yielded for the intervention on computing the SMD, Cohen’s d: −2.60 (95% CI: −4.75, −0.46; I² = 99%, *p* = 0.02) (Figure 3B).

Upon administering 300 mg of omalizumab, compared to the placebo, the mean difference for the weekly itch score was MD: −4.09 (95% CI: −4.76, −3.42) (Figure 4A). Overall, there was heterogeneity present in the findings (I2 = 89%, *p* < 0.0001) (Figure 4A). The SMD was also computed, reported as Cohen’s d: −2.21 (95% CI: −3.35, −1.06). The effect size was large and in favor of the intervention (I^2^ = 98%, *p* = 0.0002) (Figure 4B).

### 3.3. Weekly Wheal Score Outcomes

The weekly wheal scores on the administration of a 75 mg dosage of omalizumab were assessed, where the mean difference was yielded as follows: −2.36 (95% CI: −3.77, −0.94, I^2^ = 0%, *p* = 0.001) (Figure 5A). The effect direction was in favor of intervention, as reported by Cohen’s d: −0.35 (95% CI: −0.56, −0.14, I^2^ = 0%, *p* = 0.001) (Figure 5B). The effect size had a small direction in favor of 75 mg omalizumab.

On the reported wheal score outcomes post administration of 150 mg of omalizumab, the mean difference was −4.51 (95% CI: −5.94, −3.08) (Figure 6A). There was no heterogeneity (I^2^ = 0%, *p* < 0.0001). Cohen’s d had a medium effect size in favor of omalizumab intervention for CU (−0.67, 95% CI: −0.88, −0.45). Overall, there was no heterogeneity in the analysis (I^2^ = 0%, *p* < 0.0001) (Figure 6B).

The mean difference was the largest for a 300 mg intervention of omalizumab. It was computed as follows: −5.17 (95% CI: −6.9, −3.43, I^2^ = 91%, *p* < 0.0001) (Figure 7A). The strongest effect size was determined with an intervention of 300 mg omalizumab. The Cohen’s d value was reported as follows: −1.45, 95% CI: −2.2, −0.69 (Figure 7B). There was high heterogeneity (I^2^ = 96%, *p* = 0.0002).

### 3.4. Urticaria Assessment Score 7 (UAS7) Outcomes

When comparing 75 mg dosage to the placebo of omalizumab, the mean difference in the UAS7 outcome was computed as MD: −5.51 (95% CI: −9.18, −1.84) (Figure 8A). There was moderately high heterogeneity (I^2^ = 56%, *p* = 0.003). A medium effect size was determined for the intervention, and UAS7 outcomes were reported as Cohen’s d: −0.45 (95% CI: −0.77, −0.14, I^2^ = 53%, *p* = 0.004) (Figure 8B).

When comparing a 150 mg dosage of omalizumab with a placebo, the mean difference for UAS7 was the largest of the 3 dosages tested and was computed as MD: −14.52 (95% CI: −29.11, 0.06). There was high heterogeneity (I^2^ = 100%, *p* = 0.05) (Figure 9A). On computing the SMD, the values for Cohen’s d depicted a large effect size in favor of the intervention: −6.92 (95% CI: −10.38, −3.47; I^2^ = 99%, *p* < 0.0001) (Figure 9B).

The mean difference for 300 mg intervention of omalizumab was computed as follows: −6.92 (95% CI: −10.38, −3.47, I^2^ = 94%, *p* < 0.0001) (Figure 10A). The effect size was the second largest of the three dosages, and Cohen’s d was reported as follows: −2.35 (95% CI: −3.50, −1.21, I^2^ = 98%, *p* < 0.0001) (Figure 10B).

### 3.5. “Responders” to Omalizumab Treatment

On noting the “responders” to omalizumab treatment, the highest odds of responding were documented with 300 mg of intervention compared to placebo (OR = 8.65, 95% CI = 4.42, 16.93, I2 = 63%, *p* < 0.0001) (Figure 11C). This was followed by the chances of responding to 150 mg of omalizumab treatment (OR = 2.95, 95% CI = 1.88, 4.63, I^2^ = 0%, *p* < 0.0001) (Figure 11B). The responder likelihood with a 75 mg dosage was insignificant (OR = 2.13, 95% CI = 0.85, 5.33, I^2^ = 28%, *p* = 0.11) (Figure 11A).

### 3.6. Funnel Plot and Risk of Bias Synthesis

A funnel plot is depicted in Figure 12 to assess for publication bias. It may be noticed that the majority of the studies tend to fall within the vertex of the central line, meaning that while some studies may be underrepresented in the literature, there were low chances of publication bias in our meta-analysis.

When assessing the biases arising from the randomization process, 9 out of the 10 studies had low concerns, whereas 1 study had some concerns. When calibrating the biases that arose due to deviating from the intended intervention, five studies had low risks, whereas five studies had some concerns. On noting the bias arising due to missing outcome data, seven studies had low concerns, whereas three studies had some concerns. Concerning bias in the measurement of the outcomes, seven studies had low concerns, whereas three studies had some concerns. For biases in the selection of the reported result, five studies had low concerns, while five studies had some concerns. Overall, seven studies had low concerns, while three studies had some concerns (Figure 13).

## 4. Discussion

This meta-analysis analyzed the latest evidence regarding the efficacy and safety of different omalizumab dosing regimens for adult and pediatric patients with CSU across 10 RCTs. Omalizumab was more effective than placebo in improving weekly itch scores, weekly wheal scores, urticarial assessment score 7 (UAS7), and overall response rates. The effect size was the largest for a 150 mg dose of omalizumab, categorized by UAS7 scores and weekly wheal scores. A 300 mg dose of omalizumab had a more prominent improvement in weekly itch scores among patients, as well as an increased overall likelihood to complete remission. Overall, there is a strong efficacy correlation between omalizumab with both 150 and 300 mg dosing. Our meta-analysis adds to the current literature [27,28] on clinical data on the use of omalizumab for the treatment of CSU.

It is imperative to review the key time gaps between which the intervention was conducted. The XCUISITE trial administered omalizumab every 2 or 4 weeks for a total of 24 weeks, totaling 6–12 doses, whereas the ASTERIA I and II trials administered subcutaneous omalizumab every 4 weeks for a total of 24 weeks, totaling 6 doses. The GLACIAL trial also spanned 24 weeks, with subcutaneous administration every 4 weeks. The NCT01599637, MOA trial administered omalizumab every 4 weeks for a total of 12 weeks, giving a total of 3 doses. The X-ACT trial had a total administration period of 28 weeks, with 4-weekly administration totaling 7 doses. The MYSTIQUE trial administered a total of 1 dose of omalizumab and followed the patients for 24 weeks. In the trial with the following ID, NCT01713725, patients were administered omalizumab for 14 weeks, with 5 total doses. In NCT03328897, omalizumab was injected every 4 weeks for a total of 3 doses. The POLARIS trial administered omalizumab every 4 weeks for 12 weeks, totaling 3 doses.

The goal of omalizumab treatment is to rapidly ameliorate symptoms and reduce further use of medications, thereby improving the disease manifestations of CSU [29,30]. Three dosing regimens were compared for potential efficacy, and the strongest improvement in UAS7 scores and weekly itch scores were found for 150 mg, followed by 300 mg, of omalizumab Q4 weekly compared to placebo. The weekly wheal scores improvement was higher with 300 mg, and the complete resolution of symptoms, categorized as a UAS7 of 0, was 8.65× higher with 300 mg dosing of omalizumab Q4 weekly compared to placebo. Taken together, the strongest efficacy was found for both 150 and 300 mg Q4 weekly dosing. The higher effect size with the 150 mg dosing translates to excellent control of UAS7 ≤ 6 with 150 mg omalizumab. A complete response explained in treatment guidelines as “the absence of and complete protection from symptoms” was similarly measured by the UAS7 = 0 and was strongest with 300 mg omalizumab. Taken together, our findings are in line with previous meta-analyses [30,31], even after including efficacy data from the latest randomized controlled trials, favoring the use of 300 mg or 150 mg Q4 weekly omalizumab.

The UAS7 score is self-reported by the patient and is the gold standard for classifying patients as complete responders, partial responders, or non-responders [2]. UAS7 measures scores daily for seven consecutive days with two versions: once-daily documentation and the UAS7_TD_ with twice-daily documentation [32]. Both scores have high sensitivity to change and may be used to determine clinical response to treatment [33]. The UAS7 can be measured with two responder definitions, as in our study, defined as a UAS7 ≤ 6 or a UAS = 0. A UAS7 of 0 represents the most favorable outcome for the patient, as it leads to the complete resolution of symptoms. The EAACI/GA^2^LEN/EDF/WAO international urticaria guidelines recommend once-daily UAS use to determine disease activity and response to treatment [34,35]. A limitation, however, of the UAS7 score is that it may record a non-urticaria-related itch, which may have diluted the effect of omalizumab in patients who received a complete reduction of symptoms across the RCTs [20]. As in our study, while different parameters were analyzed to assess the response to omalizumab, including weekly itch scores and weekly wheal scores, the UAS7 score measures response that combines all relevant symptoms of CSU.

As 300 mg doses were found to perform slightly better than 150 mg doses in terms of overall response, it is pertinent to consider dose-response relationships. The MYSTIQUE trial [23] compared a single administration of 300 mg and 600 mg doses to establish clear dose-dependent responses; however, the trial suggested no additional benefit for the resolution of symptoms with 600 mg doses. The plateau dose may be approximately close to a 300 mg dose; however, it remains uncertain whether doses between 300–600 mg may provide any further benefit [36,37,38]. Apart from the reduction in free IgE levels, other mechanisms through which omalizumab works among patients with CSU remain unclear [13,20]. The clinical response to omalizumab is prominent much earlier than the downregulation of the IgE receptor can occur, as patients’ symptoms start resolving as early as 24–48 h [39,40]. An in vitro experiment highlights that omalizumab does not interact with mast cells or basophils through serum factors or antibodies responsible for activation. It is likely that omalizumab is acting through different mechanisms for CSU patients [40]. While links between autoimmunity and CSU have been drawn, omalizumab has strong efficacy across all types of patients with CSU [41,42,43,44]. Our data show that omalizumab efficacy in cases of CSU is found in both 150 mg and 300 mg doses. However, we cannot conclude with certainty that there is superior efficacy of higher frequency of dosing with omalizumab and treatment responses because trials administered 1–12 doses of either 150 mg or 300 mg. It may be of consideration that individual patient responses may vary on a case-by-case basis, and healthcare professionals may need to optimize dosing as per the patients’ improvement in symptoms [45,46,47].

Omalizumab must also be recognized for other inducible urticarias, including cholinergic urticaria, contact urticaria, and aquagenic urticaria [48]. A systematic review of 43 trials, case series, reports, and cohorts assessed the benefit of omalizumab among patients with inducible urticarias [49]. The evidence obtained was strongest in favor of solar urticaria, cold urticaria, and dermographism, whereas the strong body of evidence found little support for contact urticaria, aquagenic urticaria, and vibratory angioedema [49]. Overall, omalizumab has led to early control of symptoms, which is mostly seen within a 24 h period [49]. Patient groups with inducible urticaria have also obtained partial or complete relief of symptoms, along with significant improvements in quality of life [50]. Omalizumab is also reported to be well tolerated in children, with generally low adverse events [49,51].

### 4.1. Strengths and Limitations

There are certain limitations in this study. First, there may have been recall bias with UAS7 scores, as not all patients may be consistent in assessing and reporting itch severity over a 24 h period. Similarly, patients may consider either average or maximum itch when rating the severity of symptoms once daily. There may also be variations in the number of hives over 24 h, and the manner of consistency by which patients reported their total hive count may not be similar across patients. However, so long as the UAS7 instrument was used across the same patients, the scoring should remain consistent over time. Second, all the RCTs were conducted in Western populations, which excludes patients residing in other parts of the world, thereby impacting generalizability. Third, the efficacy established in such populations may be not similar for populations in non-Western regions of the world. As such, the underlying cause of CSU remains elusive. It is unclear whether real-world treatment of CSU can mimic or surpass that of what has been reported across the RCTs [52]. Fourth, the use of concomitant medications in these patients with CSU receiving omalizumab treatment is not clear, and the influence of these therapies on the overall response to omalizumab cannot be quantified or explained in our results. The strengths of this study were the methodology and design of the included studies; as such, all were RCTs and were double-blinded, thereby excluding reporting bias [53]. Further, our synthesis focused on both UAS7 scores of 0 or ≤ 6, which measures partial and complete response to treatment [54]. Finally, as our goal was to synthesize the efficacy of omalizumab and different dosing regimens, we were able to identify all outcomes of interest in this study.

### 4.2. Future Directions

There were various dosing regimens administered across the RCTs in this study. We are hopeful to gain more insight into the efficacy of omalizumab in patients with different presentations of CSU. However, the efficacy of omalizumab is also being noted in other types of CU, such as chronic inducible urticaria, which is worth exploring in prospective clinical studies [49]. Although this study was designed for understanding the best dosing regimen of omalizumab for CSU, further research may explore the mechanism of action of omalizumab on basophils and mast cell activation studies, and provide a better understanding of how omalizumab relieves symptoms of CSU [55]. Studies that focus on the mechanism and speed of anti-IgE in IgE-positive, as well as alternative mechanisms of omalizumab, can help elucidate the most effective treatment dosing regimen for patients with CSU [5,56].

## 5. Conclusions

We found excellent urticarial symptom resolution with both 300 mg and 150 mg doses of omalizumab. Improvement in UAS7 scores was more prominent with 150 mg doses, whereas the odds of complete resolution of symptoms were higher with 300 mg doses. Our results support omalizumab as an effective and safe treatment option for adult and pediatric populations who may be resistant to many therapies, including high-dose H1-antihistamines. Further studies may focus on the mechanistic actions of omalizumab beyond its action on IgE levels to completely decipher the most efficacious dosing regimen for omalizumab in CSU patients.

## Figures and Tables

**Figure 1 healthcare-10-02579-f001:**
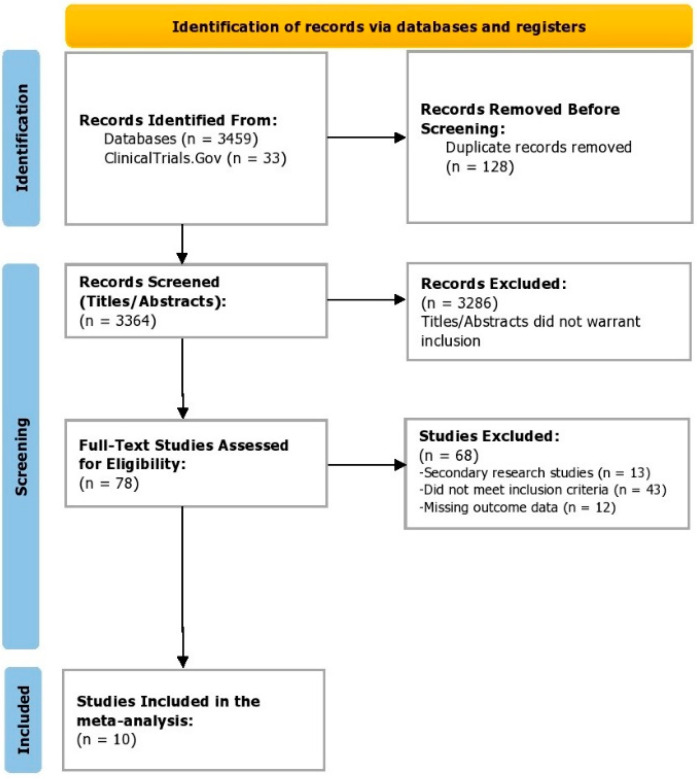
PRISMA flowchart depicting the study selection process.

**Figure 2 healthcare-10-02579-f002:**
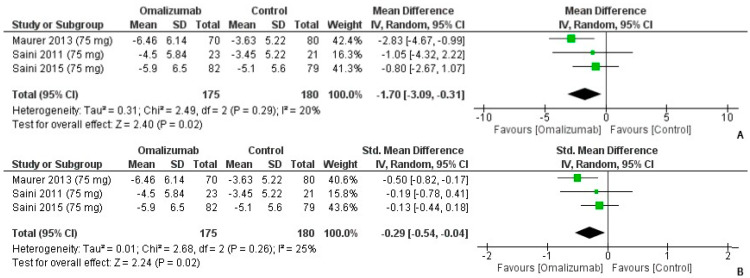
Weekly Itch Score Outcomes with 75 mg dosage of omalizumab [18,19,23]. (**A**): Mean difference: −1.70 [95% CI: −3.09, −0.31]; Heterogeneity: Tau² = 0.31; Chi² = 2.49, df = 2 (*p* = 0.29); I² = 20%; Test for overall effect: Z = 2.40 (*p* = 0.02). (**B**): Standardized mean difference: −0.29 [95% CI: −0.54, −0.04]; Heterogeneity: Tau² = 0.01; Chi² = 2.68, df = 2 (*p* = 0.26); I² = 25%; Test for overall effect: Z = 2.24 (*p* = 0.02).

**Figure 3 healthcare-10-02579-f003:**
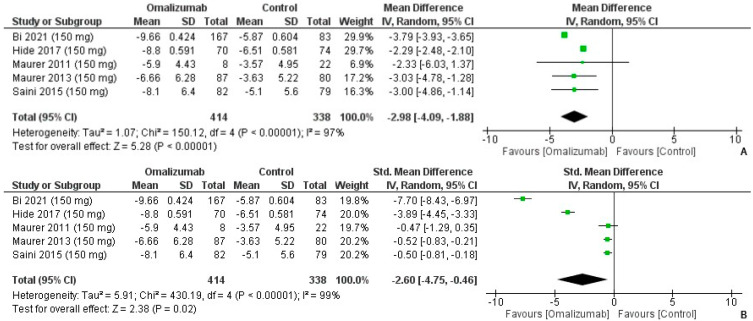
Weekly Itch Score Outcomes with 150 mg dosage of omalizumab [17,18,19,25,26]. (**A**): Mean difference: −2.98 [95% CI: −4.09, −1.88]; Heterogeneity: Tau² = 1.07; Chi² = 150.12, df = 4 (*p* < 0.00001); I² = 97%; Test for overall effect: Z = 5.28 (*p* < 0.00001). (**B**): Standardized mean difference: −2.60 [95% CI: −4.75, −0.46]; Heterogeneity: Tau² = 5.91; Chi² = 430.19, df = 4 (*p* < 0.00001); I² = 99%; Test for overall effect: Z = 2.38 (*p* = 0.02).

**Figure 4 healthcare-10-02579-f004:**
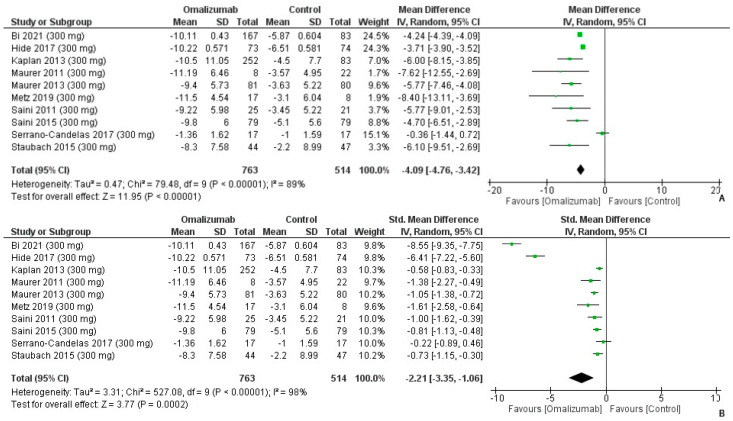
Weekly Itch Score Outcomes with 300 mg dosage of omalizumab [17,18,19,20,21,22,23,24,25,26]. (**A**): Mean difference: −4.09 [95% CI: −4.76, −3.42]; Heterogeneity: Tau² = 0.47; Chi² = 79.48, df = 9 (*p* < 0.00001); I² = 89%; Test for overall effect: Z = 11.95 (*p* < 0.00001). (**B**): Standardized mean difference: −2.21 [95% CI: −3.35, −1.06]; Heterogeneity: Tau² = 3.31; Chi² = 527.08, df = 9 (*p* < 0.00001); I² = 98%; Test for overall effect: Z = 3.77 (*p* = 0.0002).

**Figure 5 healthcare-10-02579-f005:**
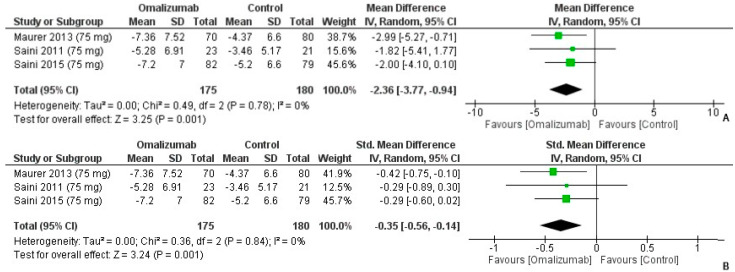
Weekly Wheal Score Outcomes with 75 mg dosage of omalizumab [18,19,23]. (**A**): Mean difference: −2.36 [95% CI: −3.77, −0.94]; Heterogeneity: Tau² = 0.00; Chi² = 0.49, df = 2 (*p* = 0.78); I² = 0%; Test for overall effect: Z = 3.25 (*p* = 0.001). (**B**): Standardized mean difference: −0.35 [95% CI: −0.56, −0.14]; Heterogeneity: Tau² = 0.00; Chi² = 0.36, df = 2 (*p* = 0.84); I² = 0%; Test for overall effect: Z = 3.24 (*p* = 0.001).

**Figure 6 healthcare-10-02579-f006:**
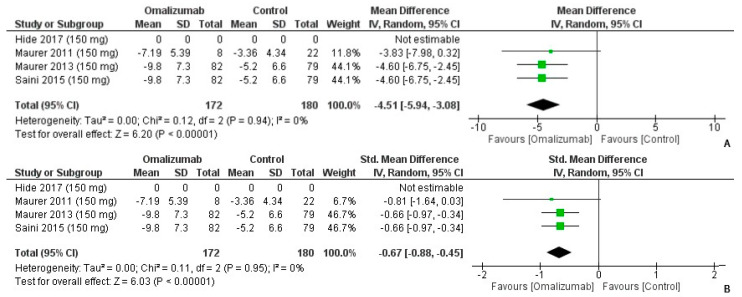
Weekly Wheal Score Outcomes with 150 mg dosage of omalizumab [17,18,19,26]. (**A**): Mean difference: −4.51 [95% CI: −5.94, −3.08]; Heterogeneity: Tau² = 0.00; Chi² = 0.12, df = 2 (*p* = 0.94); I² = 0%; Test for overall effect: Z = 6.20 (*p* < 0.00001). (**B**): Standardized mean difference: −0.67 [95% CI: −0.88, −0.45]; Heterogeneity: Tau² = 0.00; Chi² = 0.11, df = 2 (*p* = 0.95); I² = 0%; Test for overall effect: Z = 6.03 (*p* < 0.00001).

**Figure 7 healthcare-10-02579-f007:**
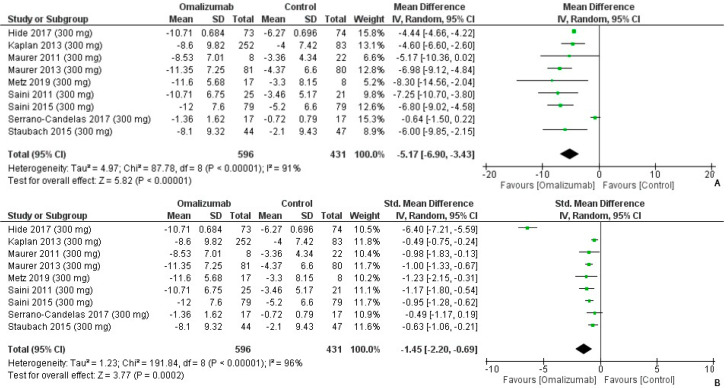
Weekly Wheal Score Outcomes with 300 mg dosage of omalizumab [17,18,19,20,21,22,23,24,26]. (**A**): Mean difference: −5.17 [95% CI: −6.90, −3.43]; Heterogeneity: Tau² = 4.97; Chi² = 87.78, df = 8 (*p* < 0.00001); I² = 91%; Test for overall effect: Z = 5.82 (*p* < 0.00001) (**B**): Standardized mean difference: −1.45 [95% CI: −2.20, −0.69]; Heterogeneity: Tau² = 1.23; Chi² = 191.84, df = 8 (*p* < 0.00001); I² = 96%; Test for overall effect: Z = 3.77 (*p* = 0.0002).

**Figure 8 healthcare-10-02579-f008:**
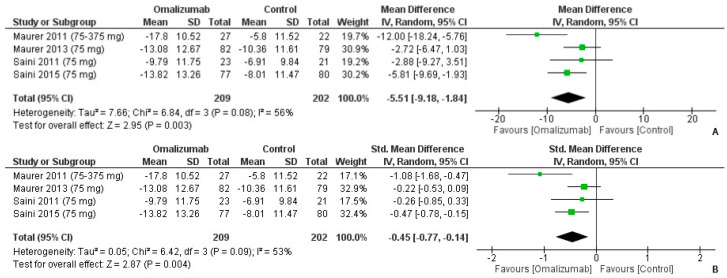
UAS7 outcomes with 75 mg dosage of omalizumab [17,18,19,23]. (**A**): Mean difference: −5.51 [95% CI: −9.18, −1.84]; Heterogeneity: Tau² = 7.66; Chi² = 6.84, df = 3 (*p* = 0.08); I² = 56%; Test for overall effect: Z = 2.95 (*p* = 0.003). (**B**): Standardized mean difference: −0.45 [95% CI: −0.77, −0.14]; Heterogeneity: Tau² = 0.05; Chi² = 6.42, df = 3 (*p* = 0.09); I² = 53%; Test for overall effect: Z = 2.87 (*p* = 0.004).

**Figure 9 healthcare-10-02579-f009:**
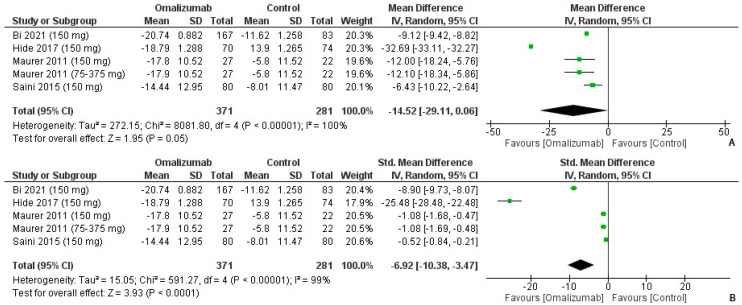
UAS7 outcomes with 150 mg dosage of omalizumab [17,18,19,25,26]. (**A**): Mean difference: −14.52 [95% CI: −29.11, 0.06]; Heterogeneity: Tau² = 272.15; Chi² = 8081.80, df = 4 (*p* < 0.00001); I² = 100%; Test for overall effect: Z = 1.95 (*p* = 0.05). (**B**): Standardized mean difference: −6.92 [95% CI: −10.38, −3.47]; Heterogeneity: Tau² = 15.05; Chi² = 591.27, df = 4 (*p* < 0.00001); I² = 99%; Test for overall effect: Z = 3.93 (*p* < 0.0001).

**Figure 10 healthcare-10-02579-f010:**
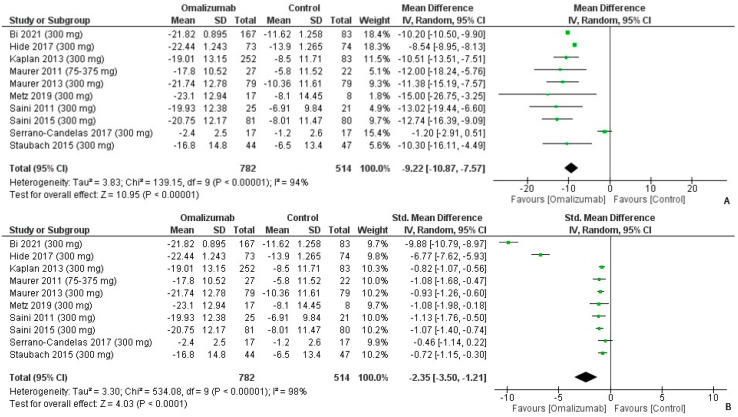
UAS7 outcomes with 300 mg dosage of omalizumab [17,18,19,20,21,22,23,24,25,26]. (**A**): Mean difference: −9.92 [95% CI: −10.87, −7.57]; Heterogeneity: Tau² = 3.83; Chi² = 139.15, df = 9 (*p* < 0.00001); I² = 94%; Test for overall effect: Z = 10.95 (*p* < 0.00001). (**B**): Standardized mean difference: −2.35 [95% CI: −3.50, −1.21]; Heterogeneity: Tau² = 3.30; Chi² = 534.08, df = 9 (*p* < 0.00001); I² = 98%; Test for overall effect: Z = 4.03 (*p* < 0.0001).

**Figure 11 healthcare-10-02579-f011:**
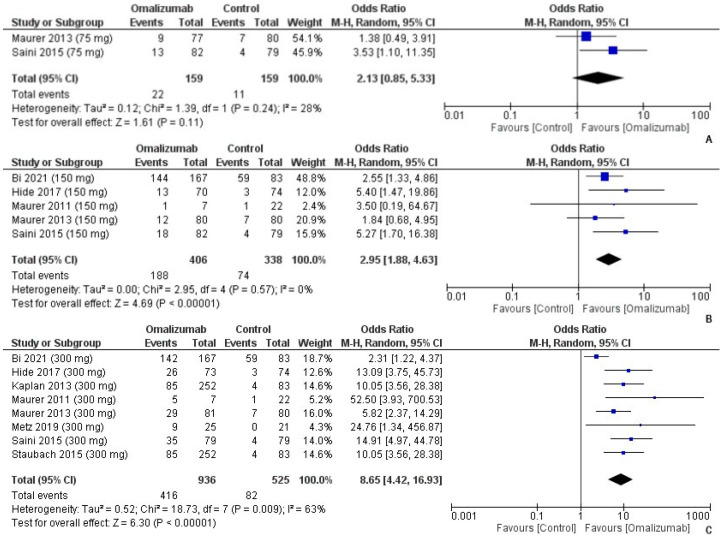
Responders to treatment with 75 mg, 150 mg, and 300 mg of omalizumab [17,18,19,20,21,22,25,26]. (**A**): 75 mg; OR = 2.13 [95% CI: 0.85, 5.33]; Heterogeneity: Tau² = 0.12; Chi² = 1.39, df = 1 (*p* = 0.24); I² = 28%; Test for overall effect: Z = 1.61 (*p* = 0.11). (**B**): 150 mg; OR = 2.95 [95% CI: 1.88, 4.63]; Heterogeneity: Tau² = 0.00; Chi² = 2.95, df = 4 (*p* = 0.57); I² = 0%; Test for overall effect: Z = 4.69 (*p* < 0.00001). (**C**): 300 mg; OR = 8.65 [95% CI: 4.42, 16.93]; Heterogeneity: Tau² = 0.52; Chi² = 18.73, df = 7 (*p* = 0.009); I² = 63%; Test for overall effect: Z = 6.30 (*p* < 0.00001).

**Figure 12 healthcare-10-02579-f012:**
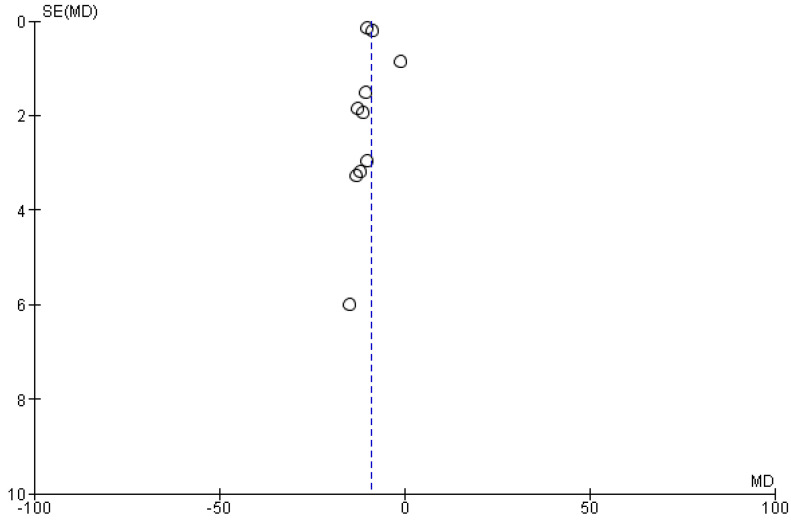
Funnel plot to visualize publication bias. The dots represent the studies distributed around the central line. These help in assessing their distribution and inspecting for publication bias.

**Figure 13 healthcare-10-02579-f013:**
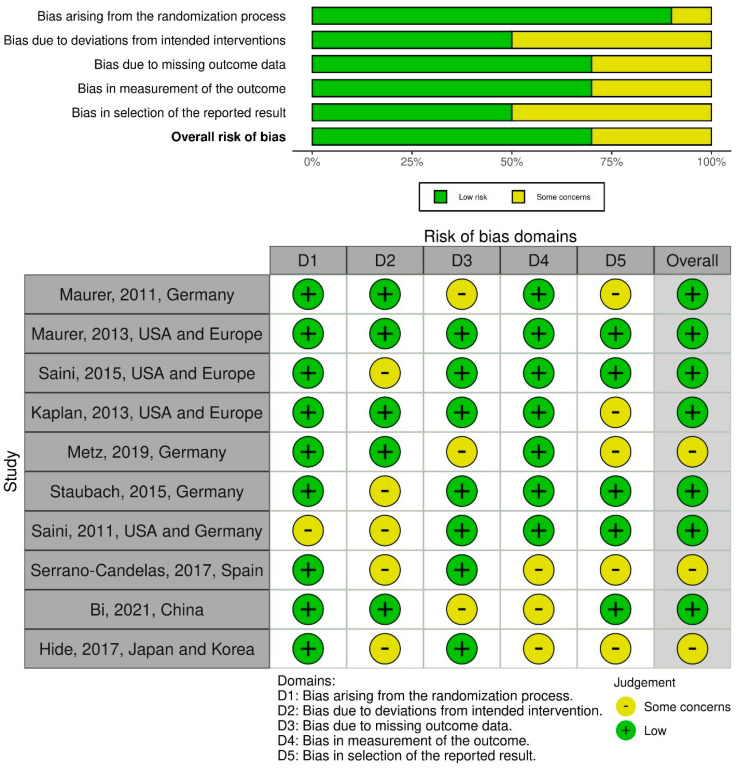
Risk of bias assessment of RCTs using the ROB-2 tool [17,18,19,20,21,22,23,24,25,26]. Weighted summary plot of the overall type of bias encountered in all studies and Traffic light plot of study-by-study bias assessment.

**Table 1 healthcare-10-02579-t001:** Characteristics of Included Studies.

ID, Name, Citation	Author, Year, Country	Dosage and Time Period	Inclusion	Mean Age	Females (IG vs. CG)	Race (IG vs. CG)
NCT00481676, XCUISITE [17]	Maurer, 2011, Germany	Omalizumab, 75 to 375 mg, subcutaneously every 2 or 4 weeks for 24 weeks for a total of 6–12 doses	Individuals between the ages of 18 and 70 years with moderate-to-severe CSU, detected with IgE autoantibodies against autoantigens, who had persistent symptoms (wheals and pruritus) despite standard antihistamine therapy ≥ 6 weeks	40.5	19/27 (70.4%) vs. 19/22 (86.4%)	All White
NCT01292473, ASTERIA I [18]	Maurer, 2013, USA and Europe	Omalizumab 75 mg or 150 mg or 300 mg, subcutaneously every 4 weeks till 12 weeks for a total of 3 doses	Individuals between the ages of 12 and 75 years with moderate-to-severe CSU, who remained symptomatic despite H1-antihistamine therapy (licensed doses) ≥ 8 weeks	42.5 ± 13.7	189/243 (77.8%) vs. 55/79 (70%)	White: 202/243 (83.1%) vs. 70/79 (89%) Non-White: 31/243 (12.8%) vs. 6/79 (8%) NA: 10/243 (4.1%) vs. 3/79 (4%)
NCT01287117, ASTERIA II [19]	Saini, 2015, USA and Europe	Omalizumab 75 mg or 150 mg or 300 mg, subcutaneously every 4 weeks till 24 weeks for a total of 6 doses	Individuals between the ages of 12–75 years with moderate-to-severe CSU who remained symptomatic despite H1-antihistamine therapy (licensed doses)	41.15	179/238 (75.2%) vs. 52/80 (65%)	White: 199/238 (83.6%) vs. 64/80 (80%) Black: 23/238 (9.7%) vs. 10/80 (12.5%) Other: 16/238 (6.7%) vs. 6/80 (7.5%)
NCT01264939, GLACIAL [20]	Kaplan, 2013, USA and Europe	Omalizumab 300 mg subcutaneously every 4 weeks till 24 weeks for a total of 6 doses	Individuals aged 12 to 75 years old with moderate-to-severe CSU; itches and hives for more than 6 consecutive weeks before enrollment despite therapy with H1-antihistamines plus H2-antihistamines, LTRAs, or both; UAS7 ≥ 16	43.1 ± 14.1	186/252 (73.8%) vs. 55/83 (66.3%)	White: 223/252 (88.5%) vs. 75/83 (90.4%)
NCT01599637, MOA [21]	Metz, 2019, Germany	Omalizumab 300 mg, subcutaneously every 4 weeks till 12 weeks for a total of 3 doses	Individuals aged 18–75 years with moderate-to-severe CSU, who remained symptomatic despite H1-antihistamine treatment at approved doses, characterized by the re-occurrence of itch and hives ≥ 6 weeks before baseline; UAS7 ≥ 16; a CSU diagnosis > 6 months; be on an approved dose of an H1-antihistamine for CSU	39.3	18/20 (90%) vs. 8/10 (80%)	All White
NCT01723072, X-ACT [22]	Staubach, 2015, Germany	Omalizumab 300 mg, subcutaneously every 4 weeks till 28 weeks for a total of 7 doses	Individuals aged 18–75 years with moderate-to-severe CSU with wheals; > 4 occurrences of angioedema in the last 6 months; symptomatic despite high-dose sg H1- antihistamine treatment (2–4 times the approved dose)	42.9 ± 12.3	30/44 (68.2%) vs. 33/47 (70.2%)	White: 42/44 (95.5%) 46/47 (97.9%) Asian: 1/44 (2.3%) vs. 1/47 (2.1%) Other: 1/44 (2.3%) vs. 0/47 (0%)
NCT00130234, MYSTIQUE [23]	Saini, 2011, USA and Germany	Omalizumab 75, 300, or 600 mg subcutaneously for a total of 1 dose and followed for 24 weeks	Individuals aged 12 to 75 years with a history of moderate-to-severe CSU ≥ 3 months (pruritus and hives for >3 days in 7 days for >6 consecutive weeks) despite treatment with an approved dose of an H1-antihistamine	40.8	44/69 (63.8%) vs. 17/21 (81%)	White: 57/69 (82.6%) vs. 18/21 (85.7%) Black/African American: 6/69 (8.7%) vs. 2/21 (9.5%) Asian: 4/69 (5.8%) vs. 1/21 (4.8%) American Indian or Alaska Native: 2/69 (2.9%) vs. 0/21 (0%)
NCT01713725 [24]	Serrano-Candelas, 2017, Spain	Omalizumab 300 mg, subcutaneously for 14 weeks, with 5 total doses	Individuals with CSU treated with omalizumab, representing a median disease duration of 6.7 years	44 ± 12.2	8/17 (47.1%) vs. 14/22 (63.6%)	NR
NCT03328897 [25]	Bi, 2021, China	Omalizumab 150 or 300 mg, injected, every 4 weeks, with 3 total doses	Children aged 6–12 years with CSU, with symptoms at least twice or 2 days per week and duration of each attack within the last 24 h	8.6	62/108 (57.4%) vs. 55/105 (52.4%)	NR
NCT02329223, POLARIS [26]	Hide, 2017, Japan and Korea	Omalizumab 150 or 300 mg subcutaneously every 4 weeks for 12 weeks, with 3 total doses	Individuals aged 12 to 75 years, with a CSU diagnosis for 6 months refractory to conventional H1AH at the time of randomization	43.57	83/144 (57.6%) vs. 48/74 (64.9%)	Japanese: 69/144 (47.9%) vs. 36/74 (48.6%) Korean: 75/144 (52.1%) vs. 38/74 (51.4%)

## Data Availability

Not applicable.

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
