# Peer review of "Efficacy of Different Dosing Regimens of IgE Targeted Biologic Omalizumab for Chronic Spontaneous Urticaria in Adult and Pediatric Populations: A Meta-Analysis"

_healthcare, 2022, doi:10.3390/healthcare10122579_

Round 1

Reviewer 1 Report

Dear authors thanks a lot for this interesting and very important publication. Indeed omalizumab is the only biologicum to treat patients with Chronic spontaneous urticaria, who have no disease-control under antihistamines.  Due to this fact we need some more informations about effectiveness and dosage.

One small recommendation 

You should mention, the need for treatment options in inducible urticaria as well. .  I recommend to publish this publication.

Author Response

Reviewer 1 Comments and Author Responses:

Comment by reviewer: 

Dear authors, thanks a lot for this interesting and very important publication. Indeed omalizumab is the only biologic to treat patients with Chronic spontaneous urticaria, who have no disease-control under antihistamines.  Due to this fact we need some more information about effectiveness and dosage.

One small recommendation 

You should mention the need for treatment options in inducible urticaria as well. I recommend to publish this publication.

Author Response: Thank you for your kind words and your input. In response to your comment, I have added a full new paragraph to focus on inducible urticaria as well. Please refer to the text highlighted in yellow in the discussion: 

“Omalizumab must also be recognized for other inducible urticarias including cho-linergic urticaria, contact urticaria, and aquagenic urticaria [47]. A systematic review of 43 trials, case series, reports and cohorts assessed the benefit of omalizumab among patients with inducible urticarias [48]. The evidence was obtained strongest in favor of solar urticaria, cold urticaria and dermographism. Whereas, the strong body of evi-dence found little support for contact urticaria, aquagenic urticaria and vibratory an-gioedema [48]. Overall, omalizumab has led to early control of symptoms, which is mostly seen within a 24-hour period [48]. Patient groups with inducible urticaria have also obtained partial or complete relief of symptoms along with significant improve-ments in quality of life [49]. Omalizumab is also reported to be well tolerated in chil-dren with generally low adverse events [48,50].”

I thank you very much for your insight and your feedback which has exponentially improved our paper.

Regards,

Dr. Zouina S.

Reviewer 2 Report

The manuscript „Efficacy of Different Dosing Regimens of IgE Targeted Biologic Omalizumab for Chronic Spontaneous Urticaria in Adult 3 and Pediatric Populations: A Meta-Analysis“  presents the comparison of the efficacy of differential dosages of omalizumab for outcomes of CSU patients (itching and wheal scores, UAS7, and response rates). In the analysis and throughout the paper, a complex and detailed statistical processing was presented/highlighted, which gave useful statistical results concerning this subject.

ABSTRACT: In the text, in the second sentence it is written: “Chronic spontaneous urticaria (CSU) is a form of the disease, which is witnessed in 2/3rd of those with chronic urticarial”. So, term „in 2/3nd“ could be replaced with more suitable words (two thirds).

After mentioning chronic spontaneous urticaria (CSU), in further text it is needed to write CSU.  Also, instead of „Omalizumab“ it is usually written „omalizumab“.

On some places in the text you write „Chronis Urticaria“ – it is usually written as „chronic urticaria“.

RESULTS: The results/characteristics of included studies are presented in the valuable table, but their results considering specific dose of omalizumab, obtained by each mentioned study, are not presented in the table.

In DISCUSSION, it could be useful to mention the influence of potential modification of period between two applications of omalizumab.

REFERENCES Please add recent reference by Zuberbier et al., 2022. (Zuberbier T, Abdul Latiff AH, Abuzakouk M, Aquilina S, Asero R, Baker D, et al. The international EAACI/GA²LEN/EuroGuiDerm/APAAACI guideline for the definition, classification, diagnosis, and management of urticaria. Allergy. 2022;77(3):734-766.)

  However, since I am a dermatologist, I can comment the text as a clinician; however, I think that it is needed to check the analyzed factors and the obtained data by a statistical professional.

Author Response

Reviewer 2 Comments and Author Responses: 

Comment 1: The manuscript „Efficacy of Different Dosing Regimens of IgE Targeted Biologic Omalizumab for Chronic Spontaneous Urticaria in Adult 3 and Pediatric Populations: A Meta-Analysis“  presents the comparison of the efficacy of differential dosages of omalizumab for outcomes of CSU patients (itching and wheal scores, UAS7, and response rates). In the analysis and throughout the paper, a complex and detailed statistical processing was presented/highlighted, which gave useful statistical results concerning this subject.

Author Response to Comment 1: Thank you for your very positive words.

Comment 2: ABSTRACT: In the text, in the second sentence it is written: “Chronic spontaneous urticaria (CSU) is a form of the disease, which is witnessed in 2/3rd of those with chronic urticarial”. So, term „in 2/3nd“ could be replaced with more suitable words (two thirds).

Author Response to Comment 2: Your insight is very helpful. It has been updated based on your suggestion. Please review the change highlighted in yellow.

Comment 3: After mentioning chronic spontaneous urticaria (CSU), in further text it is needed to write CSU.  Also, instead of „Omalizumab“ it is usually written „omalizumab“.

Author Response to Comment 3: Your insight is very helpful. It has been updated based on your suggestion. Please review the changes highlighted in yellow throughout the manuscript.

Comment 4: On some places in the text you write „Chronis Urticaria“ – it is usually written as „chronic urticaria“.

Author Response to Comment 4: Your insight is very helpful. It has been updated based on your suggestion. Please review the change highlighted in yellow.

Comment 5: RESULTS: The results/characteristics of included studies are presented in the valuable table, but their results considering specific dose of omalizumab, obtained by each mentioned study, are not presented in the table.

Author Response to Comment 5: I urge you to review the supplementary tables! All data based on dosages is there. If you have not already, kindly review Supplementary Table 2, the data sheet. It has everything you need.

Comment 6: In DISCUSSION, it could be useful to mention the influence of potential modification of period between two applications of omalizumab.

Author Response to Comment 6: I 100% agree with your comment. For this reason, I have added a new paragraph describing the various periods between applications of omalizumab. I truly hope that clears any and all issues.

“It is imperative to review the key time gaps between which the intervention was conducted. The XCUISITE trial administered omalizumab every 2 or 4 weeks for a total of 24 weeks, totaling 6-12 doses. Whereas the ASTERIA I and II trials administered subcutaneous omalizumab every 4 weeks for a total of 24 weeks totaling 6 doses. The GLACIAL trial also spanned 24 weeks with subcutaneous administration every 4 weeks. The NCT01599637, MOA trial administered omalizumab every 4 weeks for a total of 12 weeks giving a total of 3 doses. The X-ACT trial had a total administration period of 28 weeks with 4-weekly administration, totaling 7 doses. The MYSTIQUE trial administered a total of 1 dose of omalizumab and followed the patients for 24 weeks. In the trial with the following ID, NCT01713725, patients were administered omalizumab for 14 weeks with 5 total doses. In NCT03328897, omalizumab was in-jected every 4 weeks for a total of 3 doses. The POLARIS trial administered omali-zumab every 4 weeks for 12 weeks totaling 3 doses. “

Comment 7: REFERENCES Please add recent reference by Zuberbier et al., 2022. (Zuberbier T, Abdul Latiff AH, Abuzakouk M, Aquilina S, Asero R, Baker D, et al. The international EAACI/GA²LEN/EuroGuiDerm/APAAACI guideline for the definition, classification, diagnosis, and management of urticaria. Allergy. 2022;77(3):734-766.)

Author Response to Comment 7: The reference has been added. Thank you for your valuable suggestion.

Comment 8: However, since I am a dermatologist, I can comment the text as a clinician; however, I think that it is needed to check the analyzed factors and the obtained data by a statistical professional.

Author Response to Comment 8: Our co-author is a leading epidemiologist and we have another statistician/doctor in the team. We are positive our work is top of the line. 

I thank you very much for your insight and your feedback which has exponentially improved our paper.

Regards,

Dr. Zouina S.